# Thermal Stability and Purity of Graphene and Carbon Nanotubes: Key Parameters for Their Thermogravimetric Analysis (TGA)

**DOI:** 10.3390/nano14211754

**Published:** 2024-10-31

**Authors:** Markus Martincic, Stefania Sandoval, Judith Oró-Solé, Gerard Tobías-Rossell

**Affiliations:** Institut de Ciència de Materiales de Barcelona (ICMAB-CSIC), Campus de la UAB, Bellaterra, 08193 Barcelona, Spain; markus.martincic@gmail.com (M.M.); ssandoval@icmab.es (S.S.); oro@icmab.es (J.O.-S.)

**Keywords:** carbon nanotubes, graphene, thermogravimetric analysis, purification, catalyst content

## Abstract

Thermal analysis is widely employed for the characterization of nanomaterials. It encompasses a variety of techniques that allow the evaluation of the physicochemical properties of a material by monitoring its response under controlled temperature. In the case of carbon nanomaterials, such as carbon nanotubes and graphene derivatives, thermogravimetric analysis (TGA) is particularly useful to determine the quality and stability of the sample, the presence of impurities and the degree of functionalization or doping after post-synthesis treatments. Furthermore, TGA is widely used to evaluate the thermal stability against oxidation by air, which can be, for instance, enhanced by the purification of the material and by nitrogen doping, finding application in areas where a retarded combustion of the material is required. Herein, we have evaluated key parameters that play a role in the data obtained from TGA, namely, gas flow rate, sample weight and temperature rate, used during the analysis. We found out that the heating rate played the major role in the process of combustion in the presence of air, inducing an increase in the temperature at which the oxidation of CNTs starts to occur, up to ca. 100 °C (from 1 °C min^−1^ to 50 °C min^−1^). The same trend was observed for all the evaluated systems, namely N-doped CNTs, graphene produced by mechanical exfoliation and N-doped reduced graphene samples. Other aspects, like the presence of impurities or structural defects in the evaluated samples, were analyzed by TGA, highlighting the versatility and usefulness of the technique to provide information of structural aspects and properties of carbon materials. Finally, a set of TGA parameters are recommended for the analysis of carbon nanomaterials to obtain reliable data.

## 1. Introduction

Synthetic carbon allotropes, including both carbon nanotubes (CNTs) and graphene- based systems, are regarded as a robust class of materials. They were brought to the attention of the scientific community by the discovery of multi-walled and single-walled carbon nanotubes by Iijima in 1991 [1] and 1993 [2], respectively, and the first isolation and measurement of the graphene properties by Novoselov and Geim (leading to the conferring of the Novel Prize) in 2004 [3].

Carbon nanomaterials can be prepared using a variety of synthetic techniques that include arc discharge, laser ablation, chemical vapor deposition (CVD) and exfoliation of graphite. Particularly, when CVD is employed to obtain single-walled carbon nanotubes (SWCNTs), the process usually leads to the formation of impurities such as amorphous carbon and metal (catalyst residue) and graphitic nanoparticles (that can also coat the catalyst particles) or fullerenes [4]. Since the presence of these species can interfere with the properties and quality of the tubular nanostructures, their removal using a purification step, prior to their processing, is usually required. When it comes to graphene, both CVD and exfoliation of graphite remain as the most commonly employed approaches.

Due to their unique properties and versatility, carbon nanomaterials can be used by themselves or combined with other materials to improve their mechanical, chemical and electrical properties, thus creating nanocomposites [5,6,7,8,9].

The modification of the electronic structure and intrinsic properties of graphene-based materials is usually obtained via the introduction of structural defects [10,11], attaching inorganic and organic species via decoration and functionalization [12,13,14,15], and doping [16,17,18]. Meanwhile, several strategies have been reported to potentiate, modify or take advantage of the characteristics of CNTs. Due to their hollow structure, CNTs can be endohedrally modified by encapsulation of a variety of compounds within their cavities [19]. Moreover, they can undergo exohedral modification, by inducing structural doping or attaching bearing functionalities onto their walls [20,21]. This approach attracts special attention in terms of processing and improving the interaction of the material to form nanocomposites with enhanced properties [22,23]. The physical and chemical modification of carbon nanomaterials leads to materials that find application in fields that include catalysis [12], electronics [24], sensing [23] or biomedicine [25].

One particular and relatively straightforward strategy to tune the properties of carbon nanomaterials consists of the introduction of foreign atoms (chemical doping) within the honeycomb lattice that is the structural basis of both the 2D layers and the nanotube walls, respectively [26]. Nitrogen doping is the most commonly used approach for this purpose, with a myriad of reports describing synthetic methodologies to embed N atoms within the carbonaceous skeleton, thus modifying the properties of the materials [27,28,29]. The strategies employed to produce N-doped systems include the introduction of N atoms within the honeycomb lattice simultaneously with the growth of the carbonaceous material (in situ) [30] or the post-synthesis modification [31,32]. In this field, although research on doping other carbon nanostructures, like CNTs [33] and carbon nanofibers [34], is not scarce, most efforts have been focused on the structural modification of graphene-based materials, by tuning two key parameters ruling the properties of the N-doped systems, namely the concentration and nature of the N-based moieties [35,36]. N-doping allows tuning not only the conductivity [37], but also other physico-chemical properties [35], including mechanical properties [38], and thermal stability [39]. As a consequence, N-doped carbon materials find application in fields like catalysis, energy storage [34,40,41], sensors [42], electronics [43] and dye absorption.

In order to build a wide and precise portrait of the structure, composition, dimensionality and morphology, carbon nanomaterials are usually evaluated using a variety of characterization techniques. These include electron microscopy, X-ray diffraction, X-ray photoelectron, ICP, UV-Vis, FT-IR and Raman spectroscopies and thermal analysis, to name a few. Thermogravimetric analysis (TGA) is a simple, but widely used, technique to characterize carbon derivatives [44,45] and other inorganic materials, since it allows monitoring temperature-induced physical and chemical changes that occur when the system is annealed (and cooled down) under controlled conditions. By analyzing weight variations of the sample (thermal events), it is possible to determine the temperature at which a reaction takes place, and can thus provide information on the thermal stability of a material. Moreover, thermal events might offer useful information to elucidate the composition of the material and to study phenomena such as phase transitions, absorption/desorption, chemisorption, oxidation, degradation or solid–gas reactions.

The atmosphere used during the measurement plays a significant role in the information collected from the analysis [46]. When performing TGA under O_2_ or air, the transformations undergone by the sample correspond to their response towards thermal oxidation. In the case of carbon nanotubes (and other carbon-based systems) for instance, it incurs a pronounced weight loss, typically above 400 °C (temperature largely dependent on the sample) [47], as a consequence of the oxidation of the carbon skeleton into carbon dioxide, while the resulting residue (if so) might consist of oxidized inorganic materials, such as catalyst or other compounds employed for their synthesis or post-synthesis treatments [4,35]. Additionally, a weight loss, occurring at lower temperature (that in some cases can even overlap with the main thermal event), is usually attributed to the elimination of labile groups, namely water molecules, or also to bearing functionalities attached to the surface of the analyzed systems [48,49]. Under the same conditions, other nanostructures with similar surface areas or morphologies behave in a different manner, as in the case of boron nitride nanotubes, which are significantly more stable and undergo an oxidation process that leads to a weight gain at ~900 °C, corresponding to the formation of boron oxide [47]. The evaluation of similar systems in terms of morphology and dimensionality might, therefore, provide valuable information about the composition and reactivity of the sample.

When using inert gases to perform TGA (argon, helium and nitrogen—although nitrogen could be regarded as a source of N in some cases–), the weight loss, in carbon nanomaterials, corresponds to the removal of functional groups because combustion of the sp^2^ skeleton does not occur in the absence of an oxidizing ambient [50].

TGA is a widely employed and accepted technique to evaluate the behavior of carbon nanomaterials against thermal treatment. However, parameters like gas flow, heating rates and the mass of the sample, which may induce variations in the TG curve and therefore affect the interpretation of the data collected from TGA, are barely considered for the interpretation of the obtained data after analysis. Moreover, these parameters are in many cases not detailed in the scientific publications, which makes their comparison difficult. In this work, we would like to emphasize the importance of establishing appropriate parameters that can alter the output of the measurement to characterize carbon nanomaterials by TGA. We propose suitable conditions of measurement that can minimize errors when thermal analysis of carbon nanotubes and graphene derivatives is carried out. Previous studies have been devoted to analyzing the oxidation of carbon nanomaterials [47,51] and to exploring aspects like the role of the nanostructure in their thermal stability [52] or the parameters established to perform thermal analysis [53,54]. However, a detailed analysis of their influence on the study of CNTs and graphene-based materials has not been reported.

## 2. Materials and Methods

### 2.1. Chemicals

Single-walled carbon nanotubes (SWCNTs, CVD) were provided by Thomas Swan & Co., Ltd. (Consett, UK). Graphite powder (<20 µm, Buchs, Switzerland), NaNO_3_ (≥99%, Steinheim, Germany) and KMnO_4_ (99%, MO, USA) were supplied by Sigma-Aldrich, while H_2_SO_4_ (98%) and HCl (37%) were acquired from Panreac química (SLU, Darmstadt, Germany). H_2_O_2_ (35%) was obtained from Acros Organics (Geel, Belgium). NH_3_ gas (99.99%) and Ar (99.99%) were provided from Carburos Metálicos (Barcelona, Spain).

### 2.2. Methods

Purification of SWCNTs (steam and HCl treatment): In order to remove the side-product resulting from the synthesis of CNTs [55,56,57], 200 mg of SWCNTs was ground using an agate mortar and pestle. Afterwards, the sample was placed inside an open-ended silica tube and subsequently placed in a tubular furnace. The system was subsequently purged and kept under Ar atmosphere. The material was then treated with steam for 4 h, at 900 °C. After cooling down, the sample was dispersed in 200 mL of 6 M HCl, and treated overnight at 110 °C (under reflux) in order to remove the Fe catalyst used for the CNT synthesis. Samples were subsequently filtered under vacuum, washed with distilled water until neutral pH, collected and dried overnight at 60 °C. In order to evaluate the effect of the purification steps, two additional samples were considered. The first one corresponds to the SWCNTs annealed under Ar in the presence of steam (without the subsequent HCl treatment). The second sample was obtained by dispersing raw SWCNTs in 6 M HCl, followed by reflux at 110 °C, for 6 h.

Preparation of N-doped SWCNTs: Initially, to create structural defects in the SWCNTs, for the subsequent introduction of N within the graphitic network, purified SWCNTs were oxidized in the presence of HNO_3_. Briefly, 150 mg of powder was dispersed in 6 M HNO_3_ (150 mL) and subsequently refluxed at 130 °C for 45 h. After cooling down, the dispersion was filtered using a 0.2 µm PTFE membrane and washed with distilled water until reaching neutral pH. The collected powder was dried overnight at 60 °C. Afterwards, 100 mg of the obtained solid was placed inside a silica tube located in a sandwich furnace and treated for 1 h, at 500 °C, under a flow of pure ammonia gas (300 mL min^−1^ rate) [58].

Synthesis of graphene by exfoliation of graphite: Graphene was prepared using a protocol previously reported by Silva et al. [59]. Briefly, a dispersion of 60 mg of graphite powder in 200 mL of NMP was prepared by tip sonication for 30 min, using a Q700 Sonicator Qsonica (Qsonica, Newtown, CT, USA), with a power range of 20–40 W (amplitude = 50). The system was maintained inside an ice bath to avoid overheating. Afterwards, the dispersion was allowed to rest during the weekend and the supernatant (above 50% of the dispersion) was recovered.

Preparation of N-doped Reduced graphene oxide (N-doped RGO): To obtain a precursor susceptible to modification (ammonolysis reactions (N-doping)), graphene oxide (GO) was prepared by a modified Hummer’s method [35]. Briefly, concentrated H_2_SO_4_ (57.5 mL), NaNO_3_ (1.25 g) and graphite powder (2.5 g) were slowly mixed and cooled down to 0 °C; 30 min later, 7.5 g of KMnO_4_ was added, keeping the system below 20 °C. The mixture was then warmed to 35 °C and simultaneously stirred for 30 min and cooled to room temperature. Afterwards, 115 mL of water was added carefully, and the temperature of the system was fixed at 98 °C for 2 h (reflux). Finally, 500 mL of water and 2.5 mL of a 30% H_2_O_2_ solution (2.5 mL) were added. The content was cooled, centrifuged and washed with distilled water until neutral pH was reached.

N-doped RGO was synthesized by annealing GO in the presence of pure NH_3_ gas. For this purpose, the sample (100 mg) was placed inside a quartz tube located in a sandwich furnace and treated for 1 h with at the selected temperature with the gas flowing at a 300 mL min^−1^ rate.

### 2.3. Characterization

Samples were evaluated using a TA instrument TGA Q5000-IR (New Castle, DE, USA) working under a 10 mL min^−1^ balance flow of N_2_. Additional measurements were performed using a Netzsch instrument, model STA 449 F1 Jupiter^®^ (Selb, Germany). The digital resolution is in the nano-range (0.025 μg) and spans the entire measurement range (5 g). The use of the latter is specified through the manuscript. In all cases, synthetic air (80:20 N_2_:O_2_) was used as an oxidizing atmosphere. To evaluate the influence of the parameters of measurement in the combustion process, 1–20 mg of CNTs was annealed under heating rates ranging between 1 °C min^−1^ and 50 °C min^−1^. Meanwhile, the flow rates of air, ranging between 5 mL min^−1^ and 200 mL min^−1^, were employed during TG analyses. Due to the large amount of TGA performed during this work, it was not feasible to provide an error bar for each of the employed conditions. Nevertheless, error bars were included for some selected experiments by repeating the measurement three times. The aim was to provide the reader with an idea of the experimental error of this type of analysis.

The morphology of both carbon derivatives and TG residues was evaluated using a QUANTA FEI 200 FEG-ESEM (Hillsboro, OR, USA) microscope, while Energy Dispersive X-ray Spectroscopy (EDS) allowed us to determine the elemental composition of the residue obtained after the annealing treatments of SWCNTs. The diffraction pattern of the crystals, formed after the combustion of the nanotubes, was obtained from a 120 KV JEOL JEM1210 TEM (Tokyo, Japan). In the case of SWCNTs, samples for microscopy were prepared by sonicating a small amount of sample in pure ethanol. While graphite powder was dispersed in n-hexane, to evaluate the morphology of the exfoliated graphene, an aliquot of the dispersion obtained after sonication of graphite was diluted using NMP. In all cases, the obtained dispersions were placed dropwise onto lacey copper grids.

Raman spectra were acquired over the course of 15 s using a Lab Ram HR 800 Raman Jobin Yvon spectrometer (Glasgow, UK), using 532 nm and 632 nm lasers and a 50× objective. Spectra were recorded from different spots of the samples within the 100–2000 cm^−1^ range.

Elemental Analysis (EA) was performed on a Thermo Scientific™ FLASH 2000 Series CHNS Analyzer (Waltham, MA, USA) using a Mettler Toledo MX5 microbalance. To quantify the amount of catalyst present in the sample, a Superconducting Quantum Interference Device (SQUID) was employed. A Quantum design MPMS XL-7T (San Diego, CA, USA), working at 10 K (liquid nitrogen) under an external DC magnetic field, was used to obtain the hysteresis loops in the range between −50.000 and +50.000 Oe.

## 3. Results and Discussion

Due to their ease of manipulation, CNTs will be initially used to understand the role of the different TGA parameters. To complete the study, graphene-based materials will also be investigated.

The TGA curve is a useful tool to analyze the oxidation process of a material. Four main temperatures can be defined to describe the process that occurs when the sample undergoes oxidation, three of which can be determined directly from the resulting plot: *T_onset_*, *T_offset_* and *T*_50_. The extrapolated onset temperature, *T_onset_*, corresponds to the point of intersection of the starting mass baseline and the tangent to the TGA curve at the point of the maximum gradient. *T*_0_ can be obtained from the derivative thermogravimetry (DTG) curve (*dw/dT*), from which we can also calculate the surface area underneath the peak, *A*. It is worth noting that a comparison between A values is not possible for treatments performed at different heating rates, because this area is also time-dependent. The extrapolated offset temperature, *T_offset_*, is obtained by intersecting the final mass baseline at the end of analysis with the tangent to the TGA curve at the point of the maximum gradient. Finally, *T*_50_ corresponds to the temperature at which 50% of weight loss/gain occurs. Figure 1a shows the thermogravimetric curve, resulting from plotting the weight percentage (wt.%) of as-received (raw) CNTs with respect to temperature. In this case, the material is annealed under flowing air (25 mL min^−1^, 10 °C min^−1^) up to 900 °C. As it can be observed, a main thermal event occurs when annealing SWCNTs under an oxidizing atmosphere (air). The annealing treatment has induced the total combustion of the carbonaceous fraction, while a solid residue, probably resulting from the oxidation of the inorganic fraction, has been obtained and collected after cooling down.

Additionally, in order to obtain a precise overview of the material, the composition of the residue remaining from the process was assessed. In this case, SEM along with EDS analysis were carried out (Figure 1b). The obtained spectrum revealed the presence of both iron and oxygen, probably corresponding to an iron oxide, resulting from the oxidation (during the TGA) of iron catalyst, already present in the sample from the synthesis process. The structure of the residue has been determined by electron diffraction.

Figure 2 shows the morphology of the collected residue by TEM, along with its selected area electron diffraction (SAED) pattern. In both monocrystalline and polycrystalline areas, the patterns correspond to iron (III) oxide, hematite. Therefore, the presence of other iron derivatives (ex. iron carbide) or species, previously detected in CNTs, can be discarded [60].

TGA requires the definition of certain experimental parameters, such as sample mass, heating rate and gas flow rate. In order to determine their role in the resulting TGA data, several experiments were then performed. First, TGA was carried out on both raw SWCNTs and SWCNTs purified using steam followed by acid treatment. In all cases, the sample was initially allowed to stabilize at room temperature for 20 min. Afterwards, it was heated up to 120 °C, at 10 °C min^−1^. The system was isothermally maintained for 20 min, under a constant air flow (20 mL min^−1^). This step was performed to eliminate any water or volatile species from the environment that might have physisorbed onto the sample. Afterwards, the system was annealed up to 900 °C and then allowed to cool down to room temperature. For statistical purposes and to obtain reliable data, five replicas were performed for each experiment.

Initially, the influence that the employed sample mass has on the TGA curve was evaluated. Both flow rate (25 mL min^−1^) and heating rate (10 °C min^−1^) were kept invariable during the analysis. Figure 3a,b register both the thermogravimetric and the calculated derivative curves obtained after annealing 1, 2, 5, 10 and 20 mg of raw SWCNTs under air. Visual inspection of TGA curves suggests slight variations in terms of combustion temperatures when different amounts of material were employed. Let us initially focus on the temperatures at which the thermal events initiate (*T_onset_*). No major differences in *T_onset_* are observed when changing the mass of sample (from 1 to 20 mg; Figure 3c). In contrast, a more pronounced variation is perceived in the *T_offset_* values. Whereas the *T_onset_* remains barely constant in the range of 1–10 mg, a significant increase is clearly visible when using 15 mg and 20 mg of sample (reaching up to 650 °C and 681 °C, respectively). This suggests that, once the minimum temperature required for starting the oxidation is reached, the combustion of the sample already starts, but the more sample is present, the longer it takes to complete the oxidation process. This is reflected by a prolongation of the thermal event, thus inducing variations in shape of the thermogravimetric curve.

As explained above, DTG curves resulting from calculating *dw/dT* (Figure 3b) allow determining the temperature at which the weight loss rate reaches its maximum (*T*_0_, solid blue circles). However, further information can be obtained by the analysis of the DTG curve, with respect to the quality, morphologic characteristics or heterogeneity of the sample. For instance, the width of the DTG curve can be used as an indicator of material purity, where a narrower peak might correspond to the presence of a higher-purity material. It is also possible to distinguish between two or more overlapping reactions. In our study, the difference in the shape of the curves cannot be attributed to sample heterogeneity because all the analyses were carried out using the same batch of CNTs. Therefore, it only depends on the time elapsed between the start of the thermal event and the complete combustion of the sample.

The presence of inorganic impurities, in the present case of iron catalyst, can also be determined by TGA (Figure 3d) [61]. As mentioned above, annealing the sample under air induces burning of the carbon present in the sample, which is eliminated from the sample in the form of CO_2_. Therefore, since the residual weight observed after the analysis corresponds to Fe_2_O_3_ (as observed by ED), it is possible to quantitatively determine the iron content from the stoichiometry of the reaction of iron (catalyst) with O_2_ ($4Fe+3O_{2}\to 2{Fe}_{2}O_{3}$). Similar values are obtained within experimental error (error bar included for one of the analyses) regardless of the employed mass. Nevertheless, a lower iron content would seem to be present when using only 1 mg of sample. Therefore, using larger amounts of sample, whenever possible, would be desirable.

Next, the role of the gas flow rate was assessed. To minimize the experimental error that might be induced by using a low amount of sample, 5 mg of SWCNTs was employed. TGA was performed by annealing raw SWCNTs at a heating rate of 10 °C min^−1^ up to 900 °C, under flowing air at 5, 10, 25, 50, 100, 150 and 200 mL min^−1^. Figure 4a,b show the thermogravimetric curves resulting from the above-mentioned treatments, along with their DTG curves. Similarly to the previous case, the variation in the *T_onset_* for the range of gas flow rates investigated does not show significant differences, suggesting that this parameter does not play a significant role in the start of the oxidation. However, a delay in the complete oxidation of the sample (*T_offset_*) is clearly visible when using low gas flow rates, namely 5 mL min^−1^ (continuous green line) and 10 mL min^−1^ (blue dotted line). This might be induced by the lack of oxygen supply, which is crucial for the combustion of the tubular carbon nanostructures. The longer lapse of time required to finish the thermal event makes the *T_offset_* values shift towards significantly higher temperatures for the samples treated under lower flow rates (691 °C and 656 °C, for 5 mL min^−1^ and 10 mL min^−1^, respectively), as it can be appreciated in Figure 4c. As expected, wider DTG peaks can be observed from the 5 and 10 mL min^−1^ analyses (Figure 4b). In the case of the TGA at 5 mL min^−1^, a *T*_0_ of more than 30 °C is observed with respect to TGA performed at ≥25 mL min^−1^. As it can be observed in Figure 4d, non-negligible differences are observed after calculating the amount of iron catalyst present in raw SWCNTs from the different TGA curves. In particular, the determined Fe content would seem to increase in a significant manner when the analysis is carried out using a gas flow rate of 150 mL min^−1^ and there is a clear tendency to determine a much larger amount of catalyst content when the highest flow rate (200 mL min^−1^) was used. This can be explained in terms of the stability of the position of the pan (containing the sample) in the thermobalance during the analysis. When a large flow rate is used, the flowing gas may induce a slight push up of the pan, thus altering the TGA output data. Therefore, the use of such high flow rates does not seem appropriate to study this type of material.

Finally, the role played by the heating rate in the oxidation process was evaluated. For this purpose, a series of annealing treatments were performed on 5 mg of CNTs using an air flow rate of 25 mL min^−1^. The heating rates employed were 1, 5, 10, 15, 20 and 50 °C min^−1^. The resulting TGA is shown in Figure 5a. As it can be observed, these parameters turned out to play a major role and important changes can be observed, both in terms of the shape of the TGA curve and temperatures at which the oxidation occurs. Visual inspection suggests that there is a direct relationship between the heating rate and the temperature of combustion of the CNTs, with a range of *T_onset_* between 483 °C and 582 °C. Moreover, TG curves of the samples treated under both the lower (1 °C min^−1^) and the higher temperature (50 °C min^−1^) rates are clearly differentiated from those annealed under temperature rates ranging between 5 and 20 °C min^−1^. It is worth noting though that the catalyst content determined under the different heating rates was similar within experimental error (Figure 5c). According to previous reports, thermal conductivity and transition kinetics might be affected more significantly when the heating rate is modified. Instrument effects may also play a role [62,63]. In the case of the evaluated nanocarbons, one additional aspect should be considered. As mentioned before, the presence of impurities, resulting from the CVD process, alters the characteristics of the sample. For instance, the oxidation of amorphous carbon is known to occur at lower temperatures than CNTs [60]. The presence of this impurity in the raw material could account for the early combustion observed when the sample is treated at 1 °C min^−1^ (continuous green line). Once the combustion of the amorphous carbon starts to take place, the temperature of the system might locally increase because it is an exothermic process (releases energy), which might in turn result in the earlier oxidation of the nanotubes. This phenomenon becomes more visible when using the slowest heating rate. Despite a continuous increase in *T_onset_* being observed when increasing the heating rate, the width of the thermal event remains almost invariable up to 20 °C min^−1^ (*T_offset_*-*T_onset_
*~ 78 °C, Figure 5d), with symmetric DTG curves (Figure 5b) that suggest negligible differences during the oxidation process once started.

Interestingly, at the highest flow rate (50 °C min^−1^), marked differences in the shape of the TGA curve can be appreciated; these result in a large broadening of the DTG curve (*T_offset_-T_onset_
*~ 182 °C, Figure 5d). It is worth noticing that at the end of the TGA at 900 °C, the collected residue was black, indicating that the carbon fraction had not been completely oxidized, and it was necessary, only in this case, to perform the analysis up to 1000 °C to achieve a complete oxidation of the sample.

The behavior observed, in this case, can be attributed to the kinetics of the oxidation of the CNTs, which involves a series of steps to produce CO and CO_2_ [64], and that has been previously analyzed in the case of the spontaneous combustion of coals [63]. Being a nanostructure of high molecular complexity, annealing not only promotes the combustion that occurs due to the interaction of O_2_ (from the oxidizing atmosphere, air) with active sites of the nanostructure (more susceptible to oxidation), but also may induce vibrations of the species surrounding such sites or even collisions of the surrounding species with the O_2_ molecules (thus inhibiting its consumption or approach to the active sites). Finally, the energy provided to these “neighboring” species might be dissipated or released by the system towards the active site, thus interfering with the oxidation. Therefore, temperature rate does not contribute to a more efficient oxidation or to accelerating the process, and, on the contrary, can result in partial combustion of the material [63].

The capability of the thermobalance to homogeneously heat the sample can also contribute to the obtained results. Let us consider that the temperature of the furnace increases from A to B. In order to obtain appropriate and trustable data of the process, the entire sample might reach the target conditions (temperature B) homogeneously, before its weight is registered by the equipment; on the contrary, the analysis might induce deviations between the temperature of the program and the real temperature (*T_real_*) of the sample. This is the case when the system is annealed too fast and *T_real_* is lower than the expected temperature (T set in the controller). This is the case of the sample analyzed at a heating rate of 50 °C min^−1^. The slope of the TG curve also decreases due to a more prolonged time of oxidation of the sample (*T_offset_
*~ 764 °C), similarly affecting the calculation of the DTG.

By analyzing the set of parameters described above, we have clearly stressed the importance, not only of carefully selecting the conditions of the measurements in TGA, but also of always using the same parameters, to obtain suitable and comparable information from this technique. Next, to complete the study, we will analyze thermal curves resulting from the oxidation of as-received (raw) SWCNTs and SWCNTs that have undergone different post-synthesis treatments. Initially, and according to the results obtained above, we employed the following parameters to obtain reliable information from the analysis: 5 mg of sample, a heating rate of 10 °C min^−1^ and a gas flow of 25 mL min^−1^. These parameters were selected considering aspects like the shape of the TG curve (which indicates the homogeneity of the oxidation process), variations in the commencement and completion of the thermal event, and determination of the inorganic residue. Figure 6a shows both TGA and DTG curves of the as-received material and purified SWCNTs that were obtained by treating the sample under steam and hydrochloric acid (hereafter purified SWCNTs). This protocol, previously reported by Ballesteros et al., has been demonstrated to be efficient for the removal of impurities, while preventing the introduction of structural defects in the SWCNTs [57]. As consequence, the sample requires higher energy to undergo combustion, compared to the pristine ones. This agrees with the shift observed for *T*_0_ (DTG curve) in the case of the purified CNTs with respect to the raw material.

Raman spectroscopy provides information about the structure, crystallinity and purity of carbon derivatives. Two bands are commonly present in the Raman spectra of graphene derivatives (including CNTs). The first one, located around 1570 cm^−1^, corresponds to the E_2_g photon at the Brillouin zone center and it is characteristic of the vibrations of sp^2^ C atoms along the graphitic conjugated network. Meanwhile, the D band (around 1350 cm^−1^) results from stimulating sp^3^ carbon atoms, being generally attributed to the presence of defects in the structure of the carbon nanostructures. Moreover, these vibrations would also be associated with the presence of non-crystalline species in the sample, as in the case of amorphous carbon. Thus, the *I_D_/I_G_* ratio is usually employed as an indicator of the quality of the CNTs and graphene. Raman spectra, resulting from irradiation of both purified and raw CNTs with a 632 nm laser, are shown in Figure 6b. The spectra were normalized for comparison. After the purification treatment, the D band decreased (*I_D_/I_G_* ratio from 13.7 ± 0.4 to 9.8 ± 1.6 (N = 4)). Since the purification protocol involves annealing the material under a mixture of steam/argon at 900 °C, the increase in the crystallinity of the sample can be attributed either to the elimination of defects of the CNTs walls or to the removal of amorphous carbon. SQUID was also performed for monitoring the content of catalyst in the samples. In agreement with TGA, the amount of Fe after purification decreased, starting from 1.4 wt.% (3.1 emu/g) for the raw material, down to 0.5 wt.% of Fe (1.1 emu/g) for the purified CNTs. Finally, elemental analysis revealed an increase in the C content after the purification (from 91.4% for raw material to 96.9% for purified sample), as consequence of the removal of metal catalyst by the acid treatment.

The role of steam and HCl in the purification process has been well stablished. Steam, being a mild oxidant, allows the removal of amorphous carbon, also breaking the carbon layers from graphitic particles, thus exposing the metal catalyst. The latter is easily eliminated by an acid treatment. HCl is the most widely employed acid for this purpose because a priori, it should not introduce structural defects into the nanotubes. Nevertheless, to gain further insights into the effect that each of these purification steps has on the TGA curve, we also analyzed a sample of raw SWCNTs treated only with steam (orange dotted line, Appendix A) and the raw SWCNTs treated only in 6 M HCl (magenta dotted line). For comparison, the TGA of raw and purified SWCNTs is also included. To provide evidence that the analyses performed in the present study are not sample- or equipment-dependent, a new set of samples was prepared, which was in turn analyzed using another piece of TG equipment (Appendix A). In agreement with the TGA data and Figure 7, the purified sample presents a higher *T_onset_* than the pristine/raw SWCNTs. Remarkably, the highest thermal stability is observed for both samples that have been steam-treated, i.e., purified (steam + HCl) and steam. The increase in the *T_onset_* (from ~557 °C for raw SWCNTs to ~599 °C for both steam-treated samples, Appendix A), is due to the removal of the more easily oxidizable carbonaceous species (amorphous carbon) from the sample. It is also worth noting that the lowest thermal stability is observed for raw SWCNTs. This indicates that HCl removes the catalytic particles while preventing major structural damage of the SWCNTs. To gain further insights into the role of steam and HCl, Raman spectroscopy was employed (Appendix A). In agreement with the lower stability observed in the TG curve, the HCl-treated SWCNTs show the highest *I_D_/I_G_* ratio (25 ± 3), due to the presence of amorphous carbon. On the other hand, D and G bands’ intensities, appearing in the spectra of the material after being treated under steam or steam + HCl (purified), are consistent with the efficient removal of the non-crystalline fractions. Despite the fact that a more detailed characterization by Raman and complementary techniques would be needed, the collected data seem to indicate, within experimental error, that HCl treatment does not induce the creation of additional sp^3^ C (structural defects, amorphous carbon).

Both raw and purified SWCNTs were then employed to illustrate the effect of the previously discussed TGA parameters on the resulting curves. First, the amount of sample was changed. Figure 7a shows the TGA and DTG curves resulting from the analysis of 1 mg of purified SWCNTs and 20 mg of the raw material (25 mL min^−1^ gas flow and heating rate of 10 °C min^−1^). Comparison of the data with the TGA curves obtained using the same TGA parameters for both samples (Figure 6a) reveals that the increase in the number of raw nanotubes results in a delay in the *T_offset_*. This is probably because the presence of a large amount of sample requires either more energy or time for the total combustion under similar conditions. As a consequence, the determination of the thermal stability of the materials can be compromised when using different amounts of samples, as in the case of Figure 7a, where raw SWCNTs would seem to present a higher thermal stability than the purified SWCNTs, thus leading to false conclusions.

An inverse effect is observed when comparing samples treated under different heating rates, as shown in Figure 7b. If the TGA curve of purified CNTs annealed at 10 °C min^−1^ is compared with the TGA of raw material oxidized at 50 °C min^−1^ (sample mass 5 mg, gas flow 25 mL min^−1^), the slope of the TG curve varies, being more pronounced for the purified sample. When using a higher flow rate for purified CNTs (200 mL min^−1^) than for raw CNTs (25 mL min^−1^) to anneal 5 mg of sample at a heating rate of 10 °C min^−1^ (Figure 7c), the relative residue of the raw sample seems to increase, leading to an Fe content of 3.1 wt.%, which is markedly superior to the Fe calculated from the curve obtained from the experiment performed on pristine CNTs at a lower flow rate (2.3 wt.%, Figure 6a). The observed inverse quantity of catalyst among raw and purified SWCNTs would also result in false conclusions, because the amount of iron is decreased by purification, as determined by SQUID.

The analysis detailed above on the experimental parameters to be considered for thermogravimetric analyses can be extrapolated to the evaluation of other carbon nanomaterials, as per the instance of graphene and their derivatives. In order to illustrate how mechanical processing can affect the stability against oxidation by air of these materials, we prepared few-layered graphene, following a previously reported protocol, which consists on the exfoliation of graphite powder using tip sonication [59]. The treatment, performed in NMP, induced a substantial reduction in the particle size, both in terms of thickness and lateral dimensions compared to the commercial graphite (Figure 8).

These morphological variations can be directly appreciated by TGA. As it can be observed in Figure 9a, the combustion of graphite powder (performed at 10 °C min^−1^, continuous grey line) requires the application of a high thermal energy (*T_onset_* ~781 °C). The shape of the obtained TG curve (and its corresponding DTG, Figure 9b), with a pronounced slope, suggests that the starting material is relatively homogeneous, with one single thermal event associated to the combustion process. On the contrary, once the exfoliation occurs, the TG curve of few-layered graphene (dotted yellow line) undergoes important variations, due to the damage of the sheets induced by tip sonication and NMP remaining in the collected material [65]. A slight increase in the *I_D_/I_G_* ratio is observed in the Raman spectra (*I_D_/I_G_* from 0.13 ± 0.02 for graphite to 0.17 ± 0.02 of the exfoliated sample). As a result of the processing, TGA of exfoliated graphite shows a significant decrease in its thermal stability compared to graphite powder (TGA performed at 10 °C/min^−1^). Next, TGA of the exfoliated sample was performed at two additional heating rates. As in the case of SWCNTs, the temperature at which the combustion starts closely depends on the annealing conditions. Following the same trend observed above, the more rapidly the heating process occurs, the higher the temperature of oxidation is (Appendix A), with *T_onset_* of ~528 °C and ~668 °C for the samples annealed at 1 °C min^−1^ and 40 °C min^−1^, respectively.

One particular case, in which the determination of the thermal stability by TGA is especially useful, involves the evaluation of carbon nanomaterials, the structure of which has been modified via covalent or non-covalent functionalization. A variety of reports have demonstrated that, for instance, the introduction of foreign species within the sp^2^ skeleton via replacing C atoms by dopant species like boron or nitrogen not only induces changes in the electronic behavior of the material, but also affects the stability of the sample and its response against physical and chemical processes like thermal transport [66] or oxidation [36,67]. In order to gain insights about the role of the TGA conditions in the evaluation of modified nanocarbons, we have analyzed a material that we previously reported, which has demonstrated an enhanced thermal stability against the oxidation by air compared with its non-doped counterpart [39]. The sample (N-doped reduced graphene oxide, hereafter N-doped RGO) resulted from the ammonolysis treatment of graphene oxide (GO) using pure ammonia gas at 500 °C for 1 h. By performing TGA of the sample (2 mg, heating rate 10 °C min^−1^) along with other characterization techniques, we have demonstrated that the introduction of N atoms within the honeycomb lattice of graphene induced up to 162 °C increase in the *T_onset_* with respect to a sample prepared under the same experimental conditions but replacing ammonia with argon. As shown in Figure 10a, an important difference between *T_onset_* of the samples is observed in the TGA curves of both RGO and N-doped RGO, performed under the same TGA parameters. Next, we analyzed the samples using markedly different conditions. Figure 10b shows the TGA curves of RGO treated at the highest heating rate employed in this work, 50 °C min^−1^, while N-doped RGO was annealed using a heating rate much lower, 0.2 °C min^−1^. Such a low heating rate, not previously employed within this work, was used with the aim of obtaining marked differences between the analyses. Both sample weight and air flow were kept constant. Due to the long duration of the treatment performed on the N-doped sample, and considering that no thermal event would occur under 300 °C, an initial heating rate of 10 °C min^−1^ was employed up 300 °C, to reduce the total time of analysis. Then, the system was annealed at 0.2 °C min^−1^ until the sample underwent its total combustion. The sudden change in the heating rate results in an apparent increase in the sample weight. This can be considered an artifact of the measurement, likely due to the lengthy measurement, of almost one day and a half (Figure 10e represents the weight% vs. time for N-doped RGO; Figure 10d for RGO). Despite the fact that the weight% is no longer relevant and also a high residual value is obtained, the temperature at which the oxidation process takes place it is worth discussion. In the cases of both reduced GO and N-doped RGO, the temperature at which the combustion takes place agrees with the behavior of the evaluated SWCNTs. A significant increase in the heating rate (from 10 °C min^−1^ to 50 °C min^−1^) induces a shift of *T_offset_* towards higher T (from 475 °C to 502 °C for the annealed RGOs). In the case of the N-doped RGO, the use of a very slow heating rate results in a dramatic reduction in the *T_offset_*, from ~599 °C to ~462 °C. In this way, both samples, N-doped RGO and RGO, would seem to have the same thermal stability against oxidation. The use of a slow heating rate, for N-doped RGO, results in a narrower DTG curve than RGO.

Ammonolysis treatment has also been demonstrated to be an efficient strategy to introduce N heteroatoms within the SWCNT walls [58]. Purified SWCNTs were treated with nitric acid to create structural defects, active sites for the subsequent introduction of N groups by ammonia. As expected, a large increase in the *I_D_/I_G_* ratio is observed after the nitric acid (Appendix A). Although less pronounced than in the case of GO, the presence of N, embedded within the skeleton of the SWCNTs, induced variations in the TG characterization (Appendix A). For instance, the combustion of N-doped SWCNTs occurs in a longer interval of temperature than the undoped material. Finally, an increase in the TGA heating rate induces a pronounced shift of the combustion event towards higher temperatures from *T_onset_* 608 °C and *T_offset_* 702 °C at 10 °C min^−1^ to *T_onset_* ~ 632 °C and *T_offset_* ~ 833 °C at 40 °C min^−1^. This large change in the temperature range also results in important variations in the DTG curve, with the appearance of a marked shoulder above 750 °C, which could be inferred at the lower heating rate. This suggests that the thermal event involves the oxidation of species of different characteristics.

## 4. Conclusions

We have evaluated the thermal stability against oxidation in air of a series of carbon materials using TGA. We found that the modification of the heating rate played the major role in the process of combustion of CNTs in the presence of air, leading to significant variations in the temperature at which the oxidation starts to occur, from 483 °C to 582 °C for the samples annealed at 1 °C min^−1^ and 50 °C min^−1^, respectively. The duration of the processes was also affected, with differences of *T_offset_* up to 200 °C when using the lowest and highest heating rates. A similar trend was observed when TGA was performed on graphene-based materials, thus confirming the usefulness of the technique for the evaluation of different carbon nanomaterials.

Taking into account the data collected from this study, we propose the use of 2 to 10 mg of sample, a gas flow rate in the range of 25 mL min^−1^ and 100 mL min^−1^ and a heating rate of 10 °C min^−1^ for the TGA of carbon nanomaterials. Nevertheless, it is worth pointing out that these parameters are merely indicative, and might need to be adjusted depending on the TGA equipment employed, because, for instance, the configuration of some equipment allows the analysis of larger amounts of samples. Moreover, the selection of the conditions of measurement should always be accompanied by a deep knowledge of the material to be analyzed, in order to anticipate unexpected responses that can affect the measurement process.

We believe that this study, which has allowed us to establish suitable conditions to perform TGA, might also have implications for the characterization of other inorganic, organic or composite materials, always considering the specific characteristics of each sample to avoid drawing the wrong conclusions from the data obtained from the analysis.

## Figures and Tables

**Figure 1 nanomaterials-14-01754-f001:**
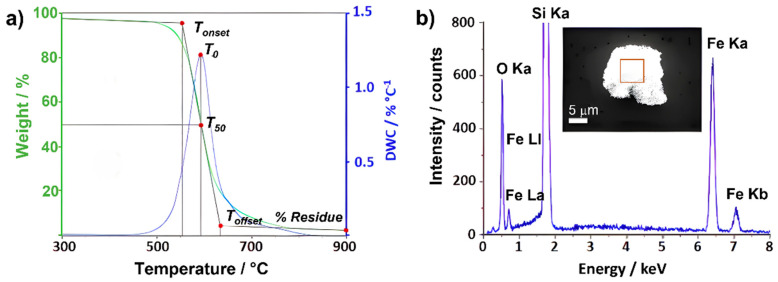
(**a**) TGA (green line) and DTG (blue line) resulting from annealing raw SWCNTs up to 900 °C (flow rate 25 mL min^−1^). (**b**) SEM micrograph (inset) and EDS resulting from analyzing a selected area (red square) of the residue collected after the annealing process. Fe and O signals are attributed to iron oxide from the oxidation of iron used as a catalyst for the growth of the SWCNTs.

**Figure 2 nanomaterials-14-01754-f002:**
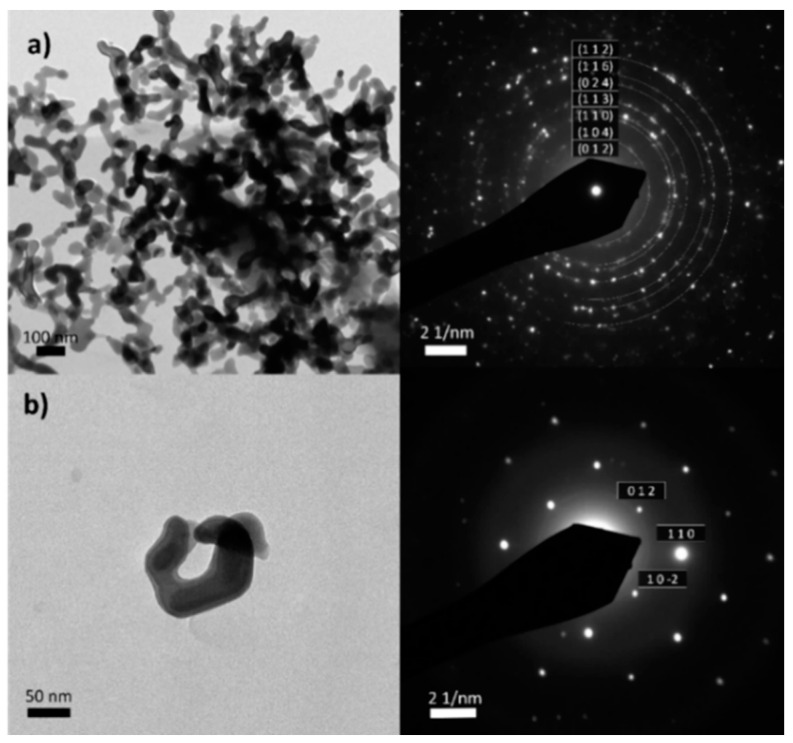
(**a**) the morphology of the collected residue by TEM. (**b**) selected area electron diffraction (SAED) pattern.

**Figure 3 nanomaterials-14-01754-f003:**
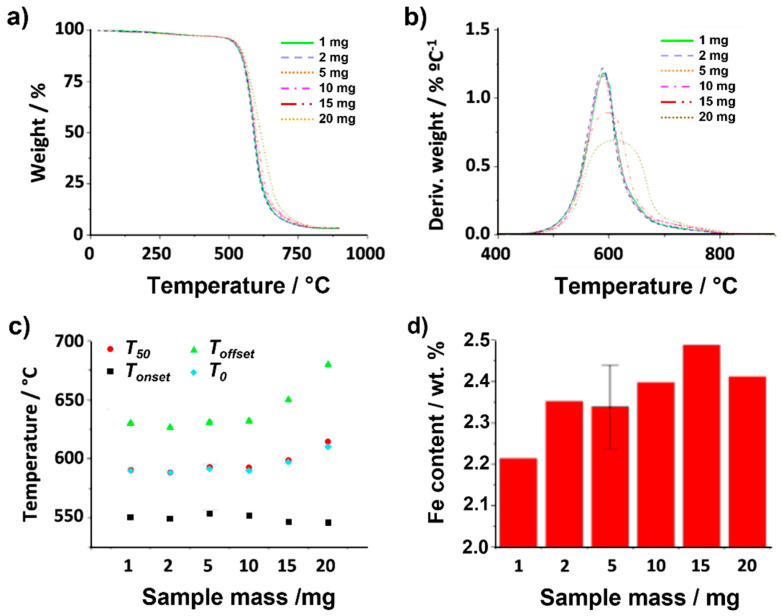
TGA of raw SWCNTs using different amounts of sample (1, 2, 5, 10, 15, 20 mg) up to 900 °C under synthetic air while keeping gas flow (25 mL min^−1^) and heating rate (10 °C min^−1^) constant. (**a**) TGA, (**b**) DTG, (**c**) temperatures determined for *T_onset_*, *T_offset_*, *T*_0_ and *T*_50_. (**d**) Iron catalyst content calculated from the collected residue after TGA of different amounts (1, 2, 5, 10, 15, 20 mg) of raw SWCNTs.

**Figure 4 nanomaterials-14-01754-f004:**
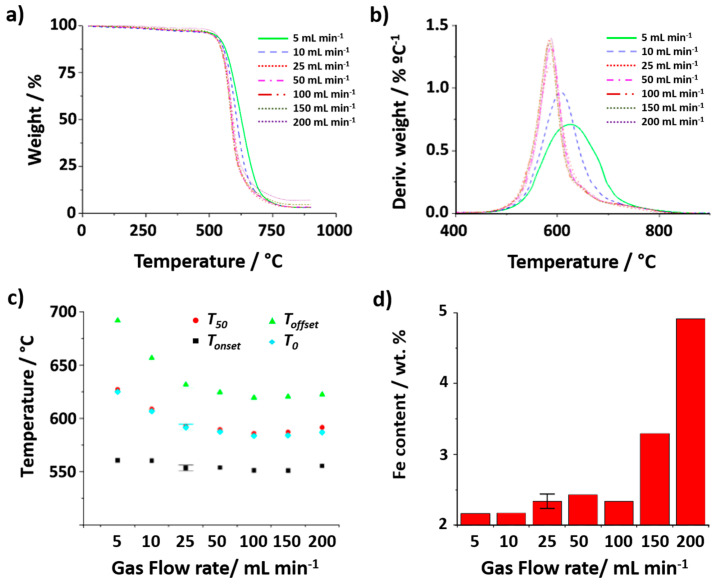
(**a**) TGA and (**b**) DTG curves, resulting from the analysis of raw SWCNTs using different flow rates (5, 10, 25, 50, 100, 150, 200 mL min^−1^) up to 900 °C under synthetic air while keeping the mass of sample (5 mg) and heating rate (10 °C min^−1^) constant. (**c**) Temperatures determined for *T_onset_*, *T_offset_*, *T*_0_ and *T*_50_. (**d**) Iron catalyst content calculated from the collected residue after TGA of 5 mg of sample using different flow rates (5, 10, 25, 50, 100, 150, 200 mL min^−1^).

**Figure 5 nanomaterials-14-01754-f005:**
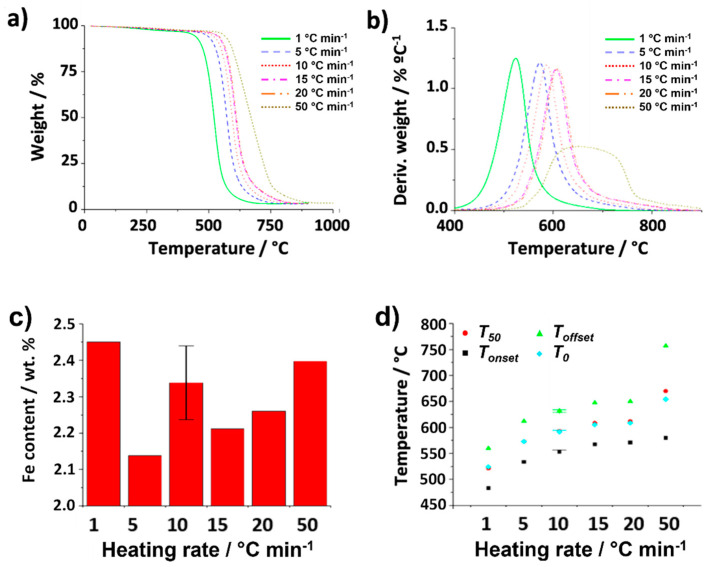
(**a**) TGA and (**b**) DTG curves obtained from analysis of raw SWCNTs using different heating rates (1, 5, 10, 15, 20, 50 °C min^−1^) up to 900 °C under synthetic air while keeping gas flow (25 mL min^−1^) and sample mass (5 mg) constant. (**c**) Iron catalyst content calculated from the collected residue after performing TGA under different heating rates (1, 5, 10, 15, 20, 50 °C min^−1^). (**d**) Temperatures determined for *T_onset_*, *T_offset_*, *T*_0_ and *T*_50_.

**Figure 6 nanomaterials-14-01754-f006:**
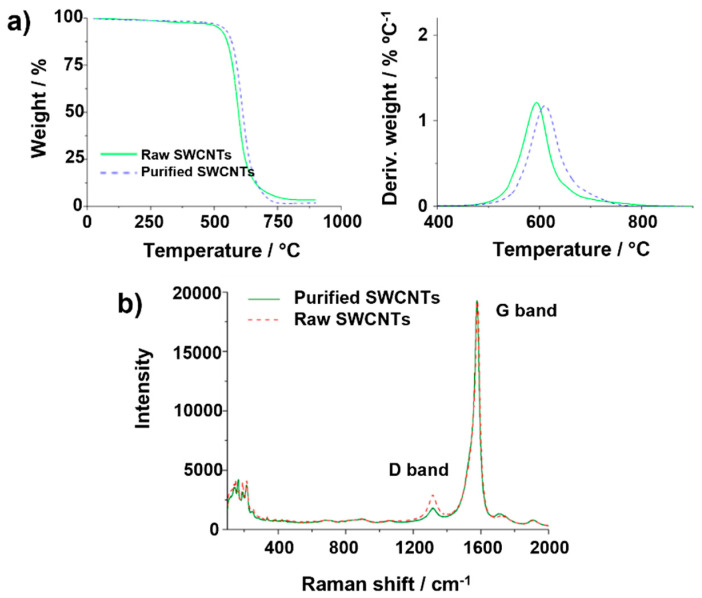
(**a**) TGA and DTG of purified and raw SWCNTs, using 5 mg of sample, a heating rate of 10 °C min^−1^ and an air gas flow of 25 mL min^−1^. (**b**) Raman spectra of raw and purified SWCNTs, obtained using a 632 nm laser.

**Figure 7 nanomaterials-14-01754-f007:**
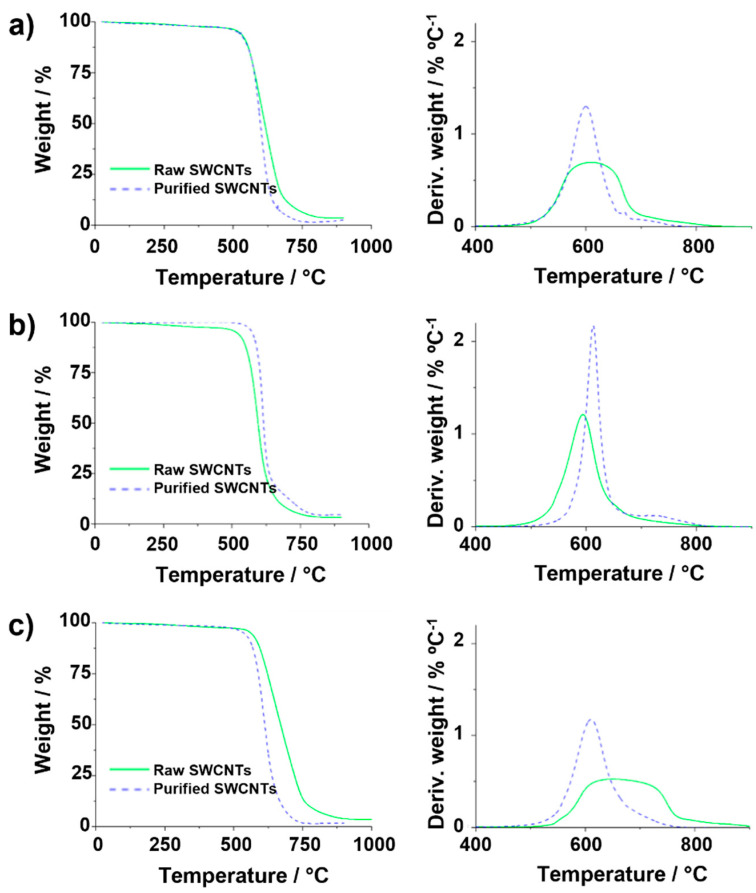
TGA and DTG of the same purified and raw SWCNTs. (**a**) Purified (1 mg) and raw (20 mg) with a heating rate of 10 °C min^−1^ and an air gas flow of 25 mL min^−1^, (**b**) TGA heating rate of 10 °C min^−1^ (purified) and 50 °C min^−1^ (raw) with 5 mg and an air gas flow of 25 mL min^−1^, (**c**) TGA gas flow of 200 mL min^−1^ (purified) and 25 mL min^−1^ (raw) using 5 mg of sample and a heating rate of 10 °C min^−1^.

**Figure 8 nanomaterials-14-01754-f008:**
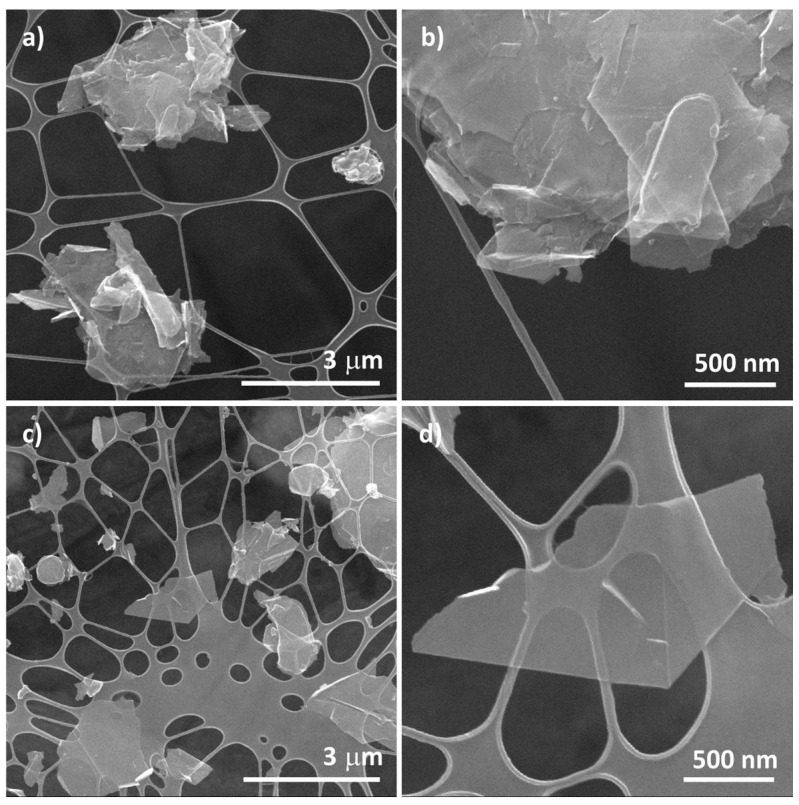
SEM images of (**a**,**b**) graphite and (**c**,**d**) few-layered graphene obtained by tip sonication of graphite powder in NMP. For ease of comparison, the same scale bars are employed for both types of material. (**b**,**d**) correspond to a magnification of a selected area in (**a**,**b**), respectively.

**Figure 9 nanomaterials-14-01754-f009:**
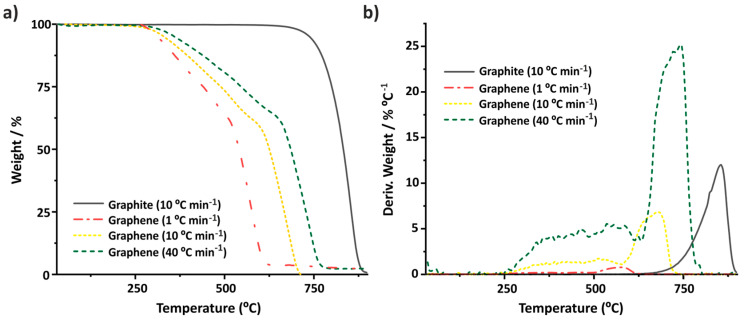
(**a**) TGA and (**b**) DTG curves of graphite powder (continuous grey line) and graphene (dotted yellow line), analyzed under flowing air at 10 °C min^−1^ and graphene (lined green line), analyzed under flowing air at 40 °C min^−1^. Graphene was obtained by tip sonication (exfoliation) of graphite. TGA was performed on a Netzsch instrument.

**Figure 10 nanomaterials-14-01754-f010:**
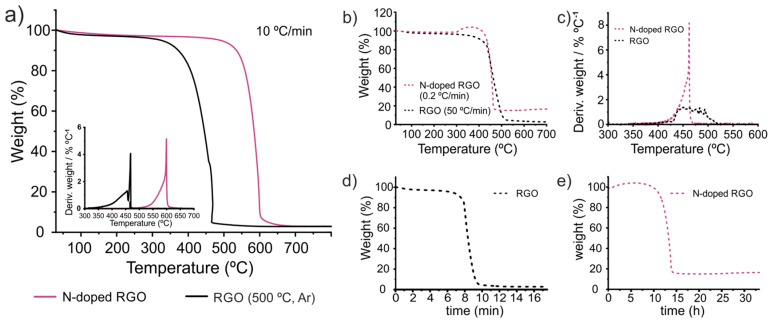
TGA curves resulting from annealing 2 mg of (**a**) RGO (continuous black line) and N-doped RGO (continuous red line) at a heating rate of 10 °C min^−1^ (the DGT curves are shown in the inset), (**b**) RGO at 50 °C min^−1^ (dotted black line) and N-doped RGO at 0.2 °C min^−1^ (dotted red line). (**c**) The DTG curves calculated from TG curves registered in (**b**). (**d**) TG curve of RGO (from treatment in (**b**)) plotted vs. the time of treatment. (**e**) TG curve of N-doped RGO (from treatment in (**b**)) plotted vs. the time of treatment. All the samples were analyzed under a constant flow of air on a Netzsch instrument, model STA 449 F1 Jupiter^®^.

## Data Availability

Data are contained within the article and Appendix A.

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
