# Peer review of "Thermal Stability and Purity of Graphene and Carbon Nanotubes: Key Parameters for Their Thermogravimetric Analysis (TGA)"

_nanomaterials, 2024, doi:10.3390/nano14211754_

Round 1
Reviewer 1 Report (Previous Reviewer 3)
Comments and Suggestions for Authors
The authors well addressed the revision requirements. The manuscript can be accepted in present form.
Author Response
Comments: The authors well addressed the revision requirements. The manuscript can be accepted in present form.
Response: We thank the reviewer for the positive comment.
Reviewer 2 Report (Previous Reviewer 4)
Comments and Suggestions for Authors
This manuscript deals with the thermal stability and purity of graphene and carbon nanotubes, and the determination of key parameters for their TGA. The paper is well-written and well-organized. However, in order for it to be considered for publication, I recommend that authors adequately address the comments I have listed below.
- Introduction: I recommend explaining the potential applications of using carbonaceous fillers (for example, producing nanocomposite, sensors, etc.). Below some references (but others could be detected by the authors) are reported:
o Journal of Science: Advanced Materials and Devices Volume 8, Issue 2, June 2023, 100557, https://doi.org/10.1016/j.jsamd.2023.100557
o Polymers, 2023, 15, 2297, https://doi.org/10.3390/polym15102297
o Nanomaterials, 2023, 13, 2653, https://doi.org/10.3390/nano13192653
o Nanomaterials 2023, 13(17), 2427; https://doi.org/10.3390/nano13172427
- Pag. 7: the authors state “Therefore, it is possible to quantitatively determine the iron present in the sample by simple stoichiometry (Figure 4a). “. I kindly suggest explaining better how this evaluation is performed.
- Pag. 8, line 313: check the spelling of “using”.
- Fig. 4: it is not clear the reason for which for a mass of “5” and a gas flow rate “25” the error bars are reported. Please, add information about the errors also for the other measurements. Check also Figure 6c, in which the error bar is reported only for one heating rate value.
- In general, the resolution of the figures should be improved, while at the same time standardizing the size of the figures and the font of the writing (which sometimes appears illegible) associated with each figure. In this regard, particular attention should be paid to figures 3,5,6,11.
- The abstract needs to be improved. Add numerical data and clarify the achievement of the research objective.
- Conclusions should be shortened and contain the most salient numerical information.
Author Response
This manuscript deals with the thermal stability and purity of graphene and carbon nanotubes, and the determination of key parameters for their TGA. The paper is well-written and well-organized. However, in order for it to be considered for publication, I recommend that authors adequately address the comments I have listed below.
Comment 1. Introduction: I recommend explaining the potential applications of using carbonaceous fillers (for example, producing nanocomposite, sensors, etc.). Below some references (but others could be detected by the authors) are reported:
o Journal of Science: Advanced Materials and Devices Volume 8, Issue 2, June 2023, 100557, https://doi.org/10.1016/j.jsamd.2023.100557
o Polymers, 2023, 15, 2297, https://doi.org/10.3390/polym15102297
o Nanomaterials, 2023, 13, 2653, https://doi.org/10.3390/nano13192653
o Nanomaterials 2023, 13(17), 2427; https://doi.org/10.3390/nano13172427
Response1 : Following the advise of the reviewer, we have highlighted the improvements induced to complex structures by filling with carbon nanomaterials and how this can potentiate their applications in diverse fields. We have include the references suggested by the referee and others we have considered relevant, as follows: “Due to their unique properties and versatility, carbon nanomaterials can be used by themselves or combined with other materials to improve their mechanical, chemical and electrical properties, thus creating nanocomposites [5-9].”
We also added “sensing” as an applicagtion in “The physical and chemical modification of carbon nanomaterials leads to materials that find application in fields that include catalysis[12], electronics[24], sensing[23] or biomedicine[25].”
Comment 2. Pag. 7: the authors state “Therefore, it is possible to quantitatively determine the iron present in the sample by simple stoichiometry (Figure 4a). “. I kindly suggest explaining better how this evaluation is performed.
Response 2: Following the suggestion of the reviewer, we have modified the paragraph including the stoichiometric reaction of the formation of Fe2O3, via oxidation of iron catalyst, in order to illustrate the calculations performed, as follows: “As it was mentioned above, annealing the sample under air induces burning of the carbon present in the sample, which is eliminated from the sample in the form of CO2. Therefore, since the residual weight observed after the analysis corresponds to Fe2O3 (as observed by ED), it is possible to quantitatively determine the iron content from the stoichiometry of the reaction of iron (catalyst) with O2 ().”
Comment 3. Pag. 8, line 313: check the spelling of “using”.
Response 3: We thank the reviewer for spotting this. It has been corrected
Comment 4. Fig. 4: it is not clear the reason for which for a mass of “5” and a gas flow rate “25” the error bars are reported. Please, add information about the errors also for the other measurements. Check also Figure 6c, in which the error bar is reported only for one heating rate value.
Response 4: We understand the concern of the reviewer, but it would be very costly and time consuming to include the error bar for all the measurements. Nevertherless, this is certinaly an important aspect that needed to be clarified in the manuscript. Therefore, the following sentence has been included: “Due to the large amount of TGA performed during this work, it was not feasible to provide an error bar for each of the employed conditions. Nevertheless, error bars were included for some selected experiments by repeating the measurement three times. The aim was to provide the reader with an idea of the experimental error of this type of analyses.”
Comment 5. In general, the resolution of the figures should be improved, while at the same time standardizing the size of the figures and the font of the writing (which sometimes appears illegible) associated with each figure. In this regard, particular attention should be paid to figures 3,5,6,11.
Response 5: We appreciate and thank to the referee for the comment. Following his/her suggestion, we have modified the figures in order to improve their quality. These modifications lead in some cases to a re-distribution of the information contained in some of the figures to be able to increase font sizes.
Comment 6. The abstract needs to be improved. Add numerical data and clarify the achievement of the research objective.
Response 6: Following referee´s suggestion, we have modified the abstract and included the most important achievement of the research.
Comment 7. Conclusions should be shortened and contain the most salient numerical information.
Response 7: The conclusions have been modified following the suggestion of the reviewer.
This manuscript is a resubmission of an earlier submission. The following is a list of the peer review reports and author responses from that submission.
Round 1
Reviewer 1 Report
Comments and Suggestions for Authors
The manuscript titled "Thermal Stability and Purity of Graphene and Carbon Nanotubes: Key Parameters for Their Thermogravimetric Analysis" by Martincic et al. presents a comprehensive study on the thermogravimetric analysis (TGA) of carbon nanomaterials. The authors provide a detailed examination of various TGA parameters and their influence on the thermal stability and purity assessment of carbon nanotubes (CNTs) and graphene derivatives (RGO and N-doped RGO). This study is highly relevant for researchers in the field of nanomaterials and offers valuable insights for optimizing TGA conditions. The manuscript could be accepted for publication after addressing the following critical points:
1. The authors should mention in the abstract the key parameters that are influencing the TGA curves.
2. In the introduction, it should be highlighted that N-doped carbons are also actively investigated in the field of electrochemical energy storage using supercapacitors; e.g. Synthesis of nitrogen-doped porous carbon nanofibers as an efficient electrode material for supercapacitors, ACS Nano (2012).
3. Recent studies related to the oxidation of CNTs should be also discussed in the introduction; e.g. "Boron nitride nanotubes versus carbon nanotubes: A thermal stability and oxidation behavior study", Nanomaterials (2020).
4. The authors state in the "Methods" section that the SWCNTs were treated with steam for 4 h at 900 °C. Since steam is considered as an activation agent for introducing nanoporosity in carbon materials, is there a possibility that they have unintentionally activated their sample, thus affecting the available surface area? Therefore, in terms of consistency, they should also thermally analyze a non-steam-treated SWCNT sample (only acid-treated).
5. Based on the previous comment, it would be also interesting to present the Raman spectrum of a SWCNT sample that has only been acid-treated (not steam-treated before).
6. The authors should highlight in the text how they have determined the onset and offset temperatures in Fig. 1a. Was this done by connecting the slopes of the horizontal and "falling" curves?
7. What is the sensitivity of the TGA balance in terms of mass? is 1 mg within the experimental error for such measurements? usually larger carbon samples of ~10 mg are employed for such purposes.
8. Please elaborate on the following statement because it is not very clear: "The use of a large flow rate may induce a slight push up of the platinum pan employed for the analysis thus altering the TGA output data".
9. The conclusions section must be expanded a bit, as it looks like the attention is given mostly on the results of the CNT samples and nothing is mentioned for the graphene-based samples.
10. Abbreviations like SWNCTs, must be defined in the beginning of the manuscript.
11. Please correct multiple typos throughout the manuscript; e.g. line 84 ("might consists"), line 116 ("during 4 h at 900 °C"), line 137 (N2 should be subscript), line 182 ("under are"), line 196 ("glass flow"), line 367 ("These to samples" and "previsouly"), line 371 ("Comparision"), line 468 ("to stablish"), etc.
Comments on the Quality of English LanguageThere are minor grammatical errors and typographical mistakes throughout the manuscript. A careful proofreading is recommended to enhance the overall quality.
Reviewer 2 Report
Comments and Suggestions for Authors
In this manuscript, the authors have evaluated aspects like amount of sample, heating rate and gas flow rate, used when TGA is performed and their effect in the combustion process of both carbon nanotubes and graphene derivatives. It is meaningful for SWCNT and graphene researchers.
However, most the measurements and data is about CNT, but the title is "Thermal stability and purity of graphene and carbon nanotubes: key parameters for their thermogravimetric analysis (TGA)" . If no more graphene data added in the revised manuscript, I suggest the author to reconsider the title.
In fact, RGO and mechanically exfoliated graphene are two typical graphene materials. The authors can try mechanically exfoliated graphene for TGA.
Reviewer 3 Report
Comments and Suggestions for Authors
The manuscript is well written and the results are well presented. However, I wonder about the real interest of this work. The authors show that TGA analysis can provide several information about the materials (in this case SWNT and RGO) which may depend on the TGA analysis parameter. This is true, but it is also well known. So what is the interest of this work. Can the authors clearly emphasize this?
As the authors write, TGA signatures are parameters (and of course materials) dependent, so they will have different effects on different materials, even SWNTs or other carbon nanomaterials which have different diameter, number of facets, nature and degree of functionalization, length, lateral size, etc. Indeed, as a function of the material, the expected information and the application, the parameters have to be carefully selected.
So contrary to what the authors conclude, the study cannot be used to establish suitable and universal conditions. Eventually they can sensible on the methodology.
Can the authors also explain the choice of these two carbon nanomaterials?
After purification of SWNT from Fe catalyst by acid treatment, introduction of oxidized groups takes place, this point is not discussed (and observed?)
N-doped RGO is analyzed, so why not N-SWNT?
I propose to publish this work after discussing the above points, but in a journal more dedicated to characterization methods.
Reviewer 4 Report
Comments and Suggestions for Authors
The manuscript, although focused on a well-trodden and popular topic in the literature, nevertheless presents several critical issues that preclude its publication in its current form.
More detailed comments are given below:
• The work is more like a technical report than an innovative research article.
• Neither the objective of the work nor the implications of the study carried out on the real applications of the carbon-based nanomaterials analyzed are clear. The following sentence “Here, we provide insights on the role of these parameters, providing tools for the selection of the optimal conditions to design reliable experiments and exploit the benefits of thermal analysis for the characterization of carbon nanomaterials.” seems very generic to me and poorly explanatory regarding the novelty of the work. In my opinion, the introduction should be rewritten by precisely defining the innovative elements of the manuscript, following a remodulation of the state of the art that takes into account only the most salient aspects and the main results, with respect to which the objective of the work is placed.
• In the text of the manuscript, the point should be written after the reference. Furthermore, consecutive references, such as [20][21][22], should be written like this [20-22].
• At the end of the introduction, the authors state: “It is worth noting that a clear and detailed description of the conditions employed for the measurements of the thermal properties of the samples allows minimizing errors when comparing analysis performed. Previous studies have been devoted to analyze the oxidation of CNTs[38] and to explore the role of the parameters established to perform thermal analysis[39][40], however, a detailed analysis of their influence on the study of CNTs and graphene- based materials has not been reported.” In this regard, the references [38], [39], [40] cited by the authors to highlight what appears to be the starting point of their study are very dated (2014, 2012, 2013). Are there really no recent works in the literature focused on the analysis of the oxidation of CNTs and the role of parameters stablished to perform thermal analysis?
•For which particular applications were the key parameters for the TGA analysis of carbon nanotubes and graphene in this work investigated?
• There is little information on the experimental part regarding the materials used. The structural parameters of Single-walled carbon nanotubes (SWCNTs, CVD) and graphite (Graphite powder) are not provided.
• The resolution and image quality are very poor. Furthermore, the writing in the figures is somewhat illegible.
• The bibliographical references are not satisfactorily updated.
• The conclusions are very vague and generic and do not present quantitative results that highlight the originality of the work.
Minor editing of English language required